# Evaluation of Intra-Abdominal Hypertension Parameters in Patients with Acute Pancreatitis

**DOI:** 10.3390/life13061227

**Published:** 2023-05-23

**Authors:** Maja Stojanović, Marko Đurić, Irina Nenadić, Nemanja Dimić, Suzana Bojić, Predrag Stevanović

**Affiliations:** 1Department of Anesthesiology and Intensive Care, University Medical Center “Zvezdara”, 11000 Belgrade, Serbia; 2Medical Faculty, University of Belgrade, 11000 Belgrade, Serbia; 3Department of Anesthesiology and Intensive Care, Clinical Center of “Dr Dragiša Mišović”, 11000 Belgrade, Serbia

**Keywords:** intra-abdominal pressure, intra-abdominal hypertension, abdominal compartment syndrome, acute pancreatitis, sepsis

## Abstract

Background: Patients with acute pancreatitis develop numerous complications and organ damage due to increased intra-abdominal pressure (IAP). These extrapancreatic complications determine the clinical outcome of the disease. Materials and methods: A total of 100 patients with acute pancreatitis were included in the prospective cohort study. Observed patients were divided into two groups according to their mean values of IAP (normal IAP values and elevated IAP values), which were compared with examined variables. Patients with intra-abdominal hypertension (IAH) were divided into four groups by IAP values, and those groups of patients were also compared with the examined variables. Results: Differences between body mass index (BMI) (*p* = 0.001), lactates (*p* = 0.006), and the Sequential Organ Failure Assessment (SOFA) score (*p* = 0.001) were statistically significant within all examined IAH groups. Differences between the mean arterial pressure (MAP) (*p* = 0.012) and filtration gradient (FG) (*p* < 0.001) were statistically significant between the first and second IAH groups in relation to the fourth. Differences in diuresis per hour (*p* = 0.022) showed statistical significance in relation to the first and third groups of IAH patients. Conclusions: Changes in IAP values lead to changes in basic vital parameters MAP, APP, FG, diuresis per hour, and lactate levels in patients with acute pancreatitis. Early recognition of changes in the SOFA score accompanying an increase in the IAP value is essential.

## 1. Introduction

Intra-abdominal hypertension (IAH) and abdominal compartment syndrome (ACS) can develop in patients with burns, sepsis, bowel obstructions, and massive fluid resuscitation after major abdominal surgery or peritonitis [1,2,3,4]. IAH also develops in patients with acute pancreatitis as a part of the pathophysiological mechanism and the occurrence of complications [5,6,7]. In 2002, the earliest written study linking acute pancreatitis to IAH indicated that elevated intra-abdominal pressure (IAP) levels could cause significant harm to organ systems and lead to high mortality rates in cases where ACS develops without signs of infection. Therefore, early abdominal decompression may be necessary [6,8]. After that, numerous other studies demonstrated the connection between IAH and acute pancreatitis, the pathophysiological mechanism of the development of IAH, its negative impact on the course of the disease, and ways to reduce the value of IAH [9,10,11,12]. IAH was shown to be a prognostic marker of the severity level of acute pancreatitis that influences further development of acute pancreatitis [13,14]. According to a report by the Abdominal Compartment Society, formerly known as the World Society of the Abdominal Compartment Syndrome (WSACS), for patients with acute pancreatitis, the incidence of IAH is 60%, and the incidence of ACS is 27% [15]. On admission, 70% of patients already have IAH, or it develops during the first few days, or the values of IAP increase [15]. A permanent presence of IAH can lead to the development of ACS and organ dysfunction, causing a high mortality rate [16]. A study conducted in a mixed ICU (medical-surgical), after the revised definitions of IAH according to the WSACS association, revealed that the prevalence of IAH was much higher than in previous studies [17]. That study showed that 30% of patients already had IAH on admission and that 15% of patients developed it within the first five days after admission to the ICU. Of patients with sepsis, 55% developed IAH and had the highest degree of mortality in relation to other risk factors (obesity and excessive fluid replacement in the first 24 h) [17]. The total prevalence was 45%, which shows that it is necessary to initially measure the IAP upon the patient’s admission to the ICU to take timely measures for its reduction and further disease progression [17,18].

Systemic microvascular dysfunction is one of the pathways leading to acute pancreatitis. Vascular leak results in marked third space fluid sequestration and may lead to ACS similar to other critical illnesses. Elevated IAP causes compression of adjacent organs and disrupts their function, leading to early impairment of organ system parameters. Alterations in the parameters of vital physiological functions can indicate the initial onset of organ dysfunction, highlighting the need for prompt therapeutic intervention. This study aimed to monitor changes in the parameters of the essential life functions within developed IAH patients suffering from acute pancreatitis.

## 2. Materials and Methods

### 2.1. Study Protocol

A prospective cohort study was conducted from January 2019 to December 2020 at the Intensive Care Unit (ICU) of “Zvezdara” University Medical Center in Belgrade. The local Medical Ethics Committee approved this study, and all patients signed informed consent forms. The study included 100 patients with acute pancreatitis. According to the revised definition [19], the patient is diagnosed with acute pancreatitis if two out of the three following criteria are present: typical abdominal pain, a threefold increase in serum amylase or lipase, and verification of findings for unclear cases of acute pancreatitis based on a computed tomography examination. This study does not include patients with acute pancreatitis who were in a chronic hemodialysis program, had a urinary catheter placed, or had surgery before admission to the ICU.

According to IAP values, patients were divided into two groups: normal IAP values (n = 40) and elevated IAP values (n = 60). According to the WSACS [1,20,21], IAH is a constant presence or repeated pathological increase in IAP ≥ 12 mmHg. Based on the WSACS recommendation and the definition, observed patients in the IAH group were divided into four groups according to the mean values of IAP and were monitored until exiting the ICU. The first group of patients had mean IAP values of 12–15 mmHg, the second group of patients had mean IAP values of 16–20 mmHg, the third group had mean IAP values of 21–25 mmHg, and the fourth group had mean IAP values of >25 mmHg. Additionally, differences in the examined variables of patients who survived or died within the IAH groups were analyzed.

### 2.2. Measuring of the Intra-Abdominal Pressure

The IAP measurement was performed according to the recommendations of WSACS [1]. All patients had a urinary catheter, with one end placed in the bladder and the other attached, by a three-way stopcock, to the urine bag and the hose of the infusion set, connected to a measuring ruler calibrated in centimeters. During the measurement, the bag was taken off and connected to the other part of the hose of the infusion set, through which 25 mL of the sterile saline solution was injected into the empty bladder. Measurements were performed on patients in the supine position, at the end of the expiratory flow, 30–60 s after solution injection. The zero point was established at the level where the midaxillary line crossed the iliac crest. Obtained values were expressed in mmHg, using a correction factor of (1 mmHg = 1.36 cmH_2_O). Measuring was conducted every twelve hours until discharge from the ICU.

The mean IAP values in all study groups were compared with the mean values of examined variables (age, body mass index (BMI), number of treatment days, Acute Physiology, Age and Chronic Health Evaluation (APACHE) II score on admission, Sequential Organ Failure Assessment (SOFA) score, values of central venous pressure (CVP), heart rate, mean arterial pressure (MAP), lactate levels, abdominal perfusion pressure (APP), filtration gradient (GF) rate, and diuresis per hour). The values of APP and FG were calculated from the obtained values of IAP according to the formulas: APP = MAP − IAP and FG = MAP − 2 × IAP.

### 2.3. Statistical Analyses

Analysis of the data was performed using the statistical data processing program IBM SPSS Statistics for Windows, Version 25.0., Armonk, NY, USA, and Statistica version 13 for Windows (StatSoft, Tulsa, OK, USA).

The mean arithmetic value, the measure of variability (standard deviation), and the minimum and maximum values of numerical variables were used within descriptive statistics. The correlation between IAP and the examined variables of the whole group, and the correlation strength, was determined by Pearson’s correlation coefficient. An ANOVA test was used to find potentially significant differences in the studied groups and changes in the examined variables occurring due to the change in IAP. Their mutual differences within the individual groups were tested by applying the post hoc Tukey test. The determination of statistical significance of the model and the influence of IAP on the variables’ changes within the whole group of patients is presented by variance analysis in the regression model. A regression analysis was performed using simple linear regression. In all applied statistical methods, the significance level was *p* < 0.05, and the high statistical significance level was *p* < 0.01. 

## 3. Results

The study included 100 patients with acute pancreatitis, of which 40 had normal IAP values and 60 had IAH. Depending on the IAP value in the group of IAH, patients were divided into four groups. The arithmetic mean value of IAP was 13.1 mmHg in the first, 17.3 mmHg in the second, 22.2 mmHg in the third, and 25.6 mmHg in the fourth group. The first group had 29, the second 20, the third 6, and the fourth 5 patients. Table 1 displays the main studied variables and their arithmetic mean values for the total number of treatment days in the ICU. Mortality in patients with acute pancreatitis in this study was 33%. In the group of patients with normal pressure, the mortality rate was 22.5% (9 deaths out of 40 patients), whereas, in the group with high levels of IAP, the mortality rate was 40% (24 deaths out of 60 patients). Comparing the IAP and treatment outcomes, we have shown that there are statistically, highly significant differences (*p* = 0.012). The examined variables were analyzed within the IAH group in survived and deceased patients; the results are shown in Table 2.

Table 3 presents the arithmetic mean values of IAP compared with the examined variables (BMI, APACHE II score on admission, MAP, APP, FG rate, diuresis per hour, lactate levels, CVP, heart rate, and the number of treatments days) within each group of IAH patients through an ANOVA test and the additional post hoc Tukey test for mutual differences, as well as the statistical significance of the variables. 

The potential associations between the mean IAP and other clinical parameters of the whole IAH group were tested by Pearson’s correlation coefficient, and their statistical significance is shown in Table 4. 

Additional analysis of obtained Pearson rank correlation coefficients showed that there is a positive and extremely strong statistical correlation between MAP and APP (r = 0.908; *p* = 0.001), between MAP and FG there is a solid and positive statistical correlation (r = 0.882; *p* = 0.001), between MAP and lactate there is a weak and negative statistical correlation (r = −0.492; *p* = 0.002), between MAP and CVP there is a weak and negative statistical correlation (r = −0.387; *p* = 0.018), between MAP and SOFA there is a negative and weak statistical correlation (r = −0.476; *p* = 0.003), between APP and FG there is an extremely strong and positive statistical correlation (r = 0.982; *p* = 0.001), between APP and CVP there is a weak and negative statistical correlation (r = −0.340; *p* = 0.040), between APP and SOFA there is a significant and negative statistical correlation (r = −0.534; *p* = 0.001), between FG and diuresis per hour there is a positive and weak statistical correlation (r = 0.33; *p* = 0.046), between FG and CVP there is a weak and negative statistical correlation (r = −0.416; *p* = 0.011), between FG and SOFA there is a significant and negative statistical correlation (r = −0.666; *p* = 0.001), between diuresis per hour and pulse there is a weak and negative statistical correlation (r = −0.344; *p* = 0.037), between diuresis per hour and SOFA there is a weak and negative statistical correlation (r = −0.483; *p* = 0.002), between lactate and SOFA there is a positive and weak statistical correlation (r = 0.432; *p* = 0.008), and between CVP and SOFA there is a positive and weak statistical correlation (r = 0.404; *p* = 0.013).

The SOFA score tendency and CVP variables tendency, as well as the regression function formula and its range for the tested variables for the entire group of IAH patients, are shown in Figure 1 and Figure 2. SOFA score and CVP values show a gradual increase in values with increasing IAP values. The fourth group of patients had the highest CVP values (>22 cmH_2_O).

## 4. Discussion

The mortality rate of patients with acute pancreatitis and the development of IAH is considerable, about 40% [5,6]. It most often occurs as a consequence of organ damage, the development of necrotic pancreatitis, and the presence of a bacterial infection. These diseased patients must be taken care of in a timely fashion to prevent possible complications, because mortality increases with the development of local or systemic complications. Together with the development of acute pancreatitis, IAH can develop as well [5,6]. The mechanism itself is complex, including increased capillary permeability caused by sepsis, hypoalbuminemia, and massive fluid resuscitation, causing retroperitoneal and visceral edema. A sudden increase in the IAP sometimes causes early organ damage and development of ACS. If ACS develops, it is usually related to the occurrence of septic shock (that does not adequately react to applied therapy), abdominal organs perfusion disorder, and dysfunction of other organs as well, as a result of mechanical influence due to increased IAP [22,23]. Previous studies demonstrate that, among patients with acute pancreatitis, 78% of patients have IAP values > 15 mmHg, while 30% have IAP values > 25 mmHg [24].

The two examined groups of patients showed a significant statistical difference in all examined variables except for age and number of days of treatment. In our study, the overall mortality was 33%. There was a significant statistical difference in the mortality rate between patients with normal and elevated IAP values; many more deceased patients were in the IAH group. There was high statistical significance in the examined variables age, APACHE II, and SOFA score, and very high statistical significance in the examined variables BMI, lactates, MAP, APP, FG, CVP, and diuresis between the group of deceased patients and survived patients in the IAH group. Only in the number of treatment days was there no statistically significant difference. Deceased patients were older and had higher values of IAP, APACHE II score, SOFA score, BMI, CVP, and lactate values and lower values of MAP, APP, FG, and diuresis per hour.

The APACHE II and SOFA scoring systems and their comparison with IAP values are often encountered in the prognostic factors determination for the development of acute pancreatitis. Previous studies showed that the high level of IAP among critically diseased patients with acute pancreatitis correlates with the level of organ damage and the length of the period spent in the ICU [25,26]. Maximal IAP value, SOFA and lactates levels, creatinine, age, APACHE II score on admission, and alkali insufficiency were significantly increased among deceased patients [25]. Within the groups of these patients, the highest mortality rate was among patients with the highest IAP values [25]. Likewise, those groups were distinguished according to SOFA level, creatinine values, ICU period, and lactates level such that they expressed higher values with the increase in the IAP. The above data show a positive correlation between the maximum values of the APACHE II and SOFA scores and the degree of increase in IAP.

A recent study showed that there is a correlation between the mean values of the APACHE II score on admission on the third and fifth days, as well as the SOFA score on the first five days of admission to the ICU and the development of IAH [27]. There is also an association between mean and maximum values of APACHE II and SOFA scores in surviving and deceased patients with IAH. However, the significance of this study is that the APACHE II score at a cut-off value of ≥12 and the SOFA score at a cut-off value of ≥6 tend to predict IAH [27].

A linear regression analysis shows the relationship between maximal IAP value and the traditional prognostic factors (APACHE II score, Ranson score on admission, and the concentration of the C reactive protein), as recently reported by Rosas et al. [13]. When the mean IAP value is determined, a significant connection appears between the APACHE II score on admission, the Imrie criterion, the Balthazar index on admission, and the number of complementary tests [13]. That study recommends using maximal IAP values to determine the severity of acute pancreatitis.

In our study, within IAH groups, the mean IAP values were compared with the mean values of the examined variables daily. The BMI variable showed a statistically significant difference with respect to the change in the IAP value within all groups. Higher BMI and body weight cause an increase in the IAP because there is a need for more strength during abdominal muscle contractions [28,29]. A previous study indicates that though obese patients have elevated IAP values, they are not in the range of IAH [30]. On the contrary, heightened IAP values were found in patients with present comorbidities who were not acutely ill. Even in these patients, the absolute values of IAP increase by 0.14 mmHg for every 1 kg/m^2^ increase in BMI [30]. 

Pathophysiological inflammatory and microcirculatory changes occur in the first two weeks of the development of acute pancreatitis, which divert the course of the disease towards one of the further disease stages, depending on the response to treatment. Depending on the body’s response to the acute illness and therapy, the number of treatment days in the ICU varies. However, in our group of patients, there were no statistically significant differences in the number of treatment days and no association between IAP and the number of days of treatment.

An analysis of the APACHE II score did not show a statistically significant difference in relation to the variable IAP, and there was also no statistically significant correlation. Within our group of patients, the results are in contrast to the previously mentioned studies, which showed that a higher APACHE II score on admission had an impact on the outcome of the disease.

The tested SOFA score variable showed a statistically significant difference within all tested groups (changes in the SOFA score of the first compared to the second group of patients, the first compared to the third and fourth groups of patients, as well as the changes of the second compared to the third and fourth groups, and the change in the SOFA score of the third group compared to the fourth). A statistically strong and positive relationship between the IAP variable and the SOFA score, as well as the change in the SOFA score due to the change in the IAP value, was also statistically proven. The increase in the IAP value was followed by an increase in the SOFA score in the entire examined group of patients. The SOFA score, as a unified indicator of several clinical indicators of the disease state, had a large and significant change within all examined groups of patients, which would additionally point to its daily evaluation and determination of disease progression, especially in patients with a constant increase in IAP.

Likewise, the IAP increase was followed by a monotonous CVP value increase in all four groups of IAH patients resulting from diaphragmatic and direct venous system compression. In particular, there was a statistically significant difference between the first and fourth groups of patients, who also had the highest values of CVP.

The MAP variable showed a statistically significant difference in relation to IAP in all examined groups and an extremely strong negative correlation with IAP, that is, as IAP values increased, MAP values decreased gradually. Elevated IAP leads to pathophysiological changes in the form of compression of the inferior vena cava and an increase in intrathoracic pressure, which further leads to reduced blood flow to the right heart and a decrease in the end-diastolic volume of the left ventricle, which results in MAP lowering.

Another study showed similar changes during the increased IAP of 12 mmHg at the pneumoperitoneum, with no change in MAP and CVP [31]. However, prolonging the operation causes a decrease in MAP and an increase in heart frequency due to developed hypotension during the surgery, which is a consequence of the prolonged influence of the high IAP [31].

The variables APP and FG showed statistically significant differences in the tested groups and a significant negative correlation with IAP. The APP changes were most considerable between the first and the third and the first and the fourth groups. The FG variable showed the largest changes between the first and second groups when compared to the third and fourth groups. According to the change in IAP, there are changes in the perfusion of all organs, especially with regard to the decrease in the perfusion of the kidneys, which react very quickly to circulatory disorders. If the compressive influence of elevated IAP is added, these changes become evident quickly. Changes in kidney perfusion are expressed by a change in diuresis per hour, which decreases evidently with an increase in the IAP value. There are statistically significant changes in the examined groups, the biggest of them being between the first and third groups of patients, probably due to targeted control therapy for reconditioning organ perfusion. Patients with IAH have impaired abdominal perfusion, which leads to APP and FG decrease.

As a result of perfusion disturbances and reduction in the excretion of toxic products, there is an accumulation of decaying substances. We found statistically, very significant differences in lactate values between the examined groups of patients, especially between the first and the fourth, the second and the fourth, and the third and the fourth groups. In addition, there was a statistically significant positive correlation between IAP and lactate changes because the patients with IAH had inadequate organ perfusion and accumulation of toxic matters due to their lowered elimination.

Additional analysis showed extremely strong and statistically significant relationships between MAP and APP, MAP and FG, and APP and FG. In addition, in patients with deterioration of their general condition, there were changes in SOFA and MAP, SOFA and APP values, as well as SOFA and diuresis per hour, and SOFA and lactate in opposite directions. Due to the inadequate response of the organism to the applied therapy, or impossibility of defending against infection, and the consequent perfusion disorder caused by the increase in IAP, there are changes in the SOFA score parameter. As a result of the overall perfusion disorder, kidney perfusion disorder manifests itself, which leads to changes in the values of APP, FG, and hourly diuresis, and the accumulation of decaying substances. The mentioned changes indicate the need for early recognition of SOFA score changes in patients with developed IAH in acute pancreatitis.

The presented research has a limitation in the number of patients studied.

## 5. Conclusions

An increase in IAP value in patients with acute pancreatitis leads to changes in the values of the MAP, APP, and FG parameters but correlations between them do not necessarily mean associations or causation. A gradual increase in the SOFA score value is the most important consideration, followed by an increase in the IAP value. The lactates level strongly depends on an IAP value increase. High IAP values lead to constant changes in the basic monitoring parameters of organ dysfunction in acute pancreatitis. To minimize complications, there is a need for timely and daily IAP measurement that aims to lower both IAP values and to optimize organ perfusion.

## Figures and Tables

**Figure 1 life-13-01227-f001:**
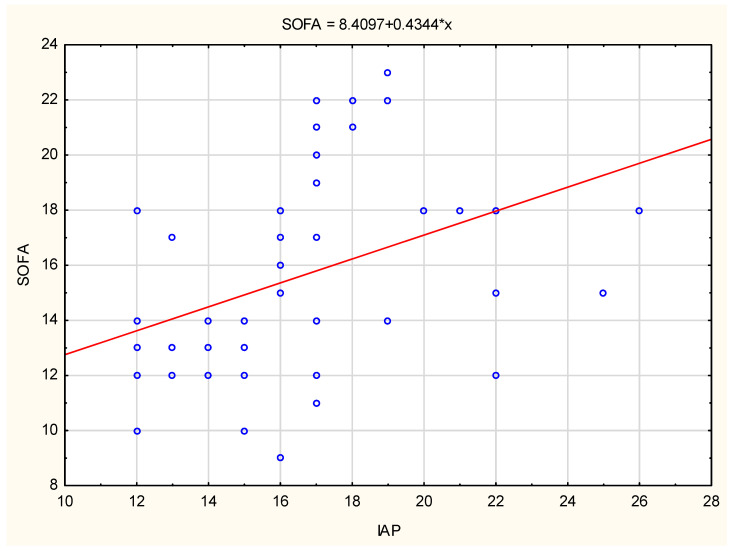
The SOFA score variation with the IAP increase. IAP—intra-abdominal pressure, SOFA—Sequential Organ Failure Assessment. *—the existence of statistical significance.

**Figure 2 life-13-01227-f002:**
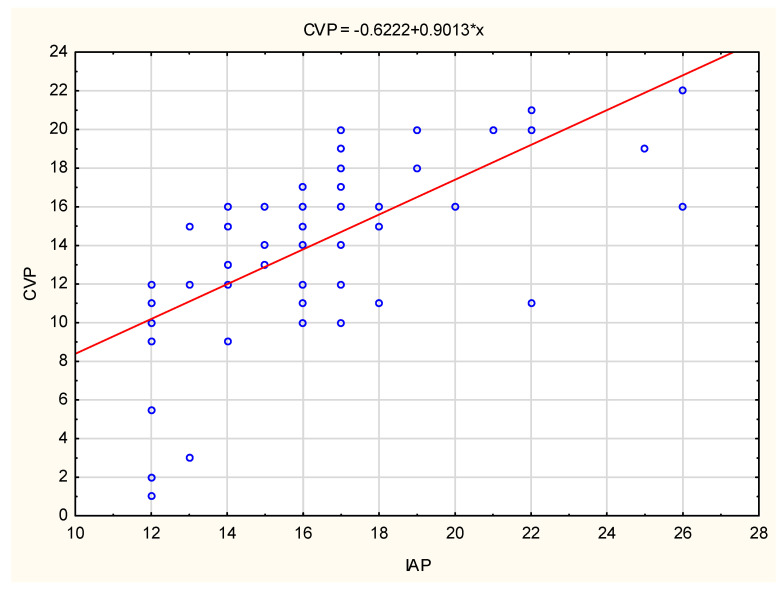
The CVP variation with the IAP increase. IAP—intra-abdominal pressure, CVP—central venous pressure. *—the existence of statistical significance.

**Table 1 life-13-01227-t001:** Anthropometric and clinical parameters of patients with normal intra-abdominal pressure and intra-abdominal hypertension.

Variables	Patients with Normal IAP (n = 40)	Patients with IAH (n = 60)	*p*-Value
IAP (mmHg)	7.6 ± 2.8	16.5 ± 4.2	0.01
Age (years)	63.5 ± 11.8	65.4 ± 12.3	0.06
Number of treatment days	5.7 ± 2.9	8.4 ± 3.4	0.19
BMI	18.7 ± 3.4	20.9 ± 4	0.001
APACHE II score	21.4 ± 6.8	27.4 ± 7.4	0.01
SOFA score	6.1 ± 1.7	8.04 ± 2.1	0.01
CVP (cmH_2_O)	8.9 ± 3.7	12.3 ± 4.7	0.002
MAP (mmHg)	87.3 ± 16.3	93.4 ± 18.5	0.001
Lactate (mmol/L)	1.1 ± 0.9	1.9 ± 1.3	0.001
APP (mmHg)	79.7 ± 16.3	78.7 ± 16.5	0.001
FG (mmHg)	72.1 ± 18.2	62.3 ± 19.3	0.005
Diuresis (mL/hour)	88.4 ± 43.9	68.3 ± 45.9	0.001

IAP—intra-abdominal pressure, IAH—intra-abdominal hypertension, BMI—Body Mass Index, APACHE II score—Acute Physiology, Age and Chronic Health Evaluation II score, SOFA—Sequential Organ Failure Assessment, CVP—central venous pressure, MAP—mean arterial pressure, APP—abdominal perfusion pressure, FG—filtration gradient.

**Table 2 life-13-01227-t002:** Examined variables of patients with intra-abdominal hypertension.

Variables	Surviving Patients (n = 36)	Deceased Patients (n = 24)	*p*-Value
IAP (mmHg)	12.4 ± 1.5	23.3 ± 2.4	0.001
Age (years)	60.2 ± 12.3	66.3 ± 12.8	0.05
Number of treatment days	5.4 ± 3.1	7.9 ± 2.9	0.09
BMI	16.3 ± 3.6	23.8 ± 4.1	0.001
APACHE II score	18.2 ± 5.9	26.2 ± 6.2	0.05
SOFA score	6.4 ± 2.1	9.3 ± 2.7	0.05
CVP (cmH_2_O)	9.3 ± 4.1	13.4 ± 4.5	0.002
MAP (mmHg)	90.4 ± 17.5	73.8 ± 19.1	0.001
Lactate (mmol/L)	1.4 ± 1.1	2.8 ± 1.3	0.001
APP (mmHg)	78.1 ± 16.1	50.5 ± 11.1	0.001
FG (mmHg)	65.7 ± 14.6	27.2 ± 8.7	0.001
Diuresis (mL/hour)	66.3 ± 26.7	38.2 ± 25.7	0.001

IAP—intra-abdominal pressure, BMI—Body Mass Index, APACHE II score—Acute Physiology, Age and Chronic Health Evaluation II score, SOFA—Sequential Organ Failure Assessment, CVP—central venous pressure, MAP—mean arterial pressure, APP—abdominal perfusion pressure, FG—filtration gradient.

**Table 3 life-13-01227-t003:** Comparison of groups of patients with intra-abdominal hypertension within the examined variables.

Tested Variables	F	*p*	Tukey HSD
IAP	143.765	0.001	IAP I–IAP II *
			IAP I–IAP III *
			IAP I–IAP IV *
BMI	88.69	0.001	BMI I–BMI II *
			BMI I–BMI III *
			BMI I–BMI IV *
			BMI II–BMI III *
			BMI II–BMI IV *
			BMI III–BMI IV *
APACHE II score	0.99	0.409	
MAP	4.272	0.012	MAP I–MAP IV *
			MAP II–MAP IV *
APP	5.217	0.005	APP I–APP III *
			APP I–APP IV *
FG	10.362	<0.001	GF I–GF III *
			GF I–GF IV *
			GF II–GF III *
			GF II–GF IV *
Diuresis per hour	3.650	0.022	Diuresis I–Diuresis III *
Lactate levels	5.027	0.006	Lactate I–Lactate IV *
			Lactate II–Lactate IV *
			Lactate III–Lactate IV *
CVP	5.53	0.003	CVP I–CVP IV *
Heart rate	2.464	0.08	
SOFA	596.155	0.001	SOFA I–SOFA II *
			SOFA I–SOFA II *
			SOFA I–SOFA IV *
			SOFA II–SOFA III *
			SOFA II–SOFA IV *
			SOFA III–SOFA IV *
Number of treatment days	2.284	0.098	

IAP—intra-abdominal pressure, BMI—Body Mass Index, APACHE II score—Acute Physiology, Age and Chronic Health Evaluation II score, MAP—mean arterial pressure, APP—abdominal perfusion pressure, FG—filtration gradient, CVP—central venous pressure, SOFA—Sequential Organ Failure Assessment, *—the existence of statistical significance.

**Table 4 life-13-01227-t004:** Correlation of intra-abdominal pressure to the tested variables.

Test Variable	r	*p*
MAP	−0.453	0.005
APP	−0.558	<0.001
FG	−0.701	<0.001
Diuresis per hour	−0.516	<0.001
Lactate levels	0.371	0.024
CVP	0.557	<0.001
SOFA	0.932	<0.001

MAP—mean arterial pressure, APP—abdominal perfusion pressure, FG—filtration gradient, CVP—central venous pressure, SOFA—Sequential Organ Failure Assessment.

## Data Availability

Available at request.

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
