# Peer review of "Evaluation of Intra-Abdominal Hypertension Parameters in Patients with Acute Pancreatitis"

_life, 2023, doi:10.3390/life13061227_

Round 1

Reviewer 1 Report

Interesting manuscript concerning an uncommon topic. The text is well structured, the results are well exposed. The references are adequate and recent. The only bias found is the low number of patients observed and the authors should point this out, however the manuscript can be accepted in this form.

Author Response

Dear reviewer, thank you for your comment. The manuscript has been corrected, there is certainly a limitation due to the small number of patients

Reviewer 2 Report

Major comments

 The research methodology (comparative statistics) is not well done. Namely, as far as can be understood, the authors wanted to compare (or examine the impact of) the mean values of IAP with clinical parameters (BMI, number of days of treatment, APACHE II score...). The authors divided the subjects into 4 groups according to the (arithmetic?) mean IAP during the stay. The author in Table 2 reported a multitude of regression equations for the particular variables for the groups. The presented methodology and obtained results do not seem to be correct for achieving the proclaimed aim of the research. Namely, in order to examine the influence of mean IAP on clinical parameters, it is necessary to create a new Table 2 (the first 2 rows of existing Table 2 are correct, but an N-number of participants-in the group must be added) instead of the existing Table 2. In this new Table 2, the arithmetic values of all variables for the particular group (including the mean IAP for each group) should be presented, with the differences between the groups tested with an ANOVA test (possibly using a posthoc test for additional mutual differences).

Furthermore, the potential associations between the mean IAP and other clinical parameters should be tested by correlation (Pearson or Spearman, depending on the test of normality of distribution, and this should be stated) for ALL subjects (not within an individual group!). Namely, some groups have too few subjects to be possible to use regression or construct a graph with a linear regression equation. Accordingly, Figure 3 should be omitted.

The existing graphs of linear regression (Figures 1 and 2) seem correctly made, but it is not clearly stated whether the regression applies to the entire cohort (N)?

 Furthermore, in addition to the currently used variables, CRP and creatinine should also be added, and perhaps other laboratory variables such as bilirubin and transaminases (laboratory variables and scores at the beginning of the hospital stay) and the CT severity score after 48-72 h from the start of the stay.

Accordingly, based on this partly wrong research methodology, the conclusions obtained in the Discussion and Conclusion sections may not be correct, so they need to be changed after the data has been correctly processed.

Minor comments

A plagiarism check of the article revealed, in subsection Measuring of the intra-abdominal pressure of section Material and Methods, nearly 100% similarity with previously published material (a similar article from the same group and the doctoral thesis of the first authors). This subsection should be substantially amended and changed. A high percent of similarity with previous material is also demonstrated in Statistical analyses section.

 Line 70-75: The inclusion and exclusion criteria for participants must be more clearly described. It was not clearly stated which patients were excluded from the study.

 Line 79-84: The division into groups therefore depends on the mean IAP (is it the arithmetic mean of all values?)

If authors decide to divide subjects into groups based on mean IAP, it should be stated clearly throughout the text. (for example: line 81-83: „The first group of patients was with the mean IAP values of 12- 15 mmHg, the second group of patients was with the mean IAP values of 16-20 mmHg, the third group was with the mean IAP values of 21-25 mmHg and fourth group was with the mean IAP values of > 25 mmHg“)

Section statistical analysis merits some textual improvements:

-          Line 104-106: „(Statistical Package for Social Sciences) version 23 (SPSS Inc., Chicago, Ill., USA)“: the name of company is IBM (and accordingly should be cited as „IBM SPSS Statistics for Windows, Version 23.0., Armonk, NY, United States” or “IBM Corp. Released 2015. IBM SPSS Statistics for Windows, Version 23.0. Armonk, NY: IBM Corp.“)

-          Line 106: „statistica“ should be written as „Statistica version 13 for Windows (StatSoft, Tulsa, OK, USA)

-          Line 108: „arithmetic middle“ must be changed to „arithmetic mean“

-          Instead the term „characteristics“ must be written as „variables“

-          Significance must be written as P (not p) across the manuscript

-          Normality distribution of the variables must be tested (for example Shapiro-Wilk or Kolmogorov-Smirnov test)

Results:

-          Line 118: Are the values in Table 1 at the time of admission? It should be stated clearly for a portion of the variables presented.

Abstract:

 - line 14-18: “ Observed 14 patients were divided into four groups according to the mean values of IAP……….“  It is necessary to clearly define the groups because the “second” and “third” groups are mentioned in the continuation of the abstract, despite their unclear definition.

Author Response

Dear reviewer, the text has been corrected according to your instructions. The statistical method has been corrected. A new statistical method, ANOVA, was developed, according to which the results were presented and the discussion and conclusion were corrected accordingly. At the same time, table number 2 was replaced with a new table and the results were supplemented with new statistical processing. Figure 3 has been removed. Figures 1 and 2 refer to the entire group of patients and this is explained in the text. Since the SOFA score was used, CRP, creatinine values and changes according to CT were not monitored, therefore they are not shown.

The IAP measurement method shown in the methodology is the method normally recommended and used in my hospital. that method is otherwise recommended by the World Association of Compartment Syndrome and according to those guidelines is used in all hospitals in order to be uniform and not to get different results due to the wrong way of measuring, in the wrong position of the patient or with a different amount and type of fluid that is administered uses, therefore there is a similarity with other conducted studies.

The patients who did not participate in the study are precisely listed

Arithmetic means of all investigated variables are listed

The exact name of the statistical program is given

„arithmetic middle“ is changed to „arithmetic mean“

The term „characteristics“ is written as „variables“

Significance is written as P across the manuscript

The mean values of the variables for the entire treatment period are shown in Table 1

The abstract has been changed in accordance with the new results

Reviewer 3 Report

I would like to thank you for inviting me to review this paper. The subject is current and vital to clinical practice. I would like to congratulate the authors for prospectively enrolling these patients. I have several issues to address:

1. The abstract should include more mathematical data instead of descriptions like "very high at the beginning" or "constant fall".

2. There are 60 patients included in this two-year study. However, this is a subset of patients; please add data regarding patients from the ICU with acute pancreatitis (AP) without intra-abdominal hypertension (IAH) and the rest of the patients that were admitted to the hospital without ICU transfer. In this way, a comparison for age, gender, BMI, medical comorbidities, and etiology becomes more meaningful.

3. Please add the interval between hospital admission and ICU transfer for each group. Are there any differences between groups? Do we assist to "too late diagnosis" phenomenon leading to significant morbidity? Do the authors plead for IAP screening for patients with acute pancreatitis?

4. Table 2 contains data regarding the trend of the examined variables. One should acknowledge the fact that the numbers in all groups (especially groups 3 and 4) are too low for a proper statistical analysis. The authors generated the regression model for variables inside the groups but did not provide a comparison between groups. An easier way to follow Table 2 would be to provide the median and IQR (most probable that the distribution is non-parametric).

5. Table 2 should include data regarding age, medical comorbidities (Charlson score), etiology, SOFA score, SIRS and organ failure (number and type).

6. How many patients in each group ended up in percutaneous or surgical procedures? What was the mortality rate in each group?

7. The authors should include a "limitations of the study" section.

Author Response

Dear reviewer, considering that one of the reviewer demanded the use of a new statistical method, a large part of the manuscript was modified according to that method. ANOVA analysis was used and according to it the results shown in Tables 2 and 3 were obtained, and the discussion and conclusion were also modified accordingly. The results of the statistical method used are presented in the abstract. No comparison was made between patients with IAH and those without IAH. The number of days of treatment outside the ICU is not shown because all patients with acute pancreatitis are admitted to the ICU in our hospital. Unfortunately, in this study, the treatment of the patients, whether surgical or the use of drugs, was not monitored. Certainly, the study has limitations, primarily due to the number of patients examined.

Round 2

Reviewer 2 Report

The authors have significantly improved the text and the results are presented much more clearly.

However, some suggestions from my first review were not accepted, which I regret to confirm.

They did not take into account my advice of additionally variables to be included (CT severity score). Also, authors did not tested normality of distribution for numerical variables (Shapiro-Wilk or Kolmogorov-Smirnov test). Finally, number of participants was not added in table 2.

Despite the above-mentioned deficiencies, the article is acceptable for publication if some additional small changes are made, which I mention in the rest of this report:

·       Line 4: the name of the first author listed as „Maja Zoran Stojanović “should be checked: might should be written “Maja Stojanovic” or “Zoran Stojanovic”?   Please check!!

·       Lines 15-19: Result subsection of the Abstract section is not well written: „Body mass index (BMI), (P = 0.001), lactates (P = 0.006), and Sequential Organ Failure Assessment Score (SOFA) score, (P = 0.001) are statistically significant within all examined groups. The mean arterial pressure (MAP), (P = 0.012) and filtration gradient (GF), (P =0.0001) are statistically significant between the first and the second group of patients in relation to the fourth. Diuresis per hour (P = 0.022) shows statistical significance in relation to the first and third groups of patients.“

It should be rephrased as „Differences between body mass index (BMI), (P = 0.001), lactates (P = 0.006), and the Sequential Organ Failure Assessment Score (SOFA) score, (P = 0.001) were statistically significant within all examined groups. Differences between the mean arterial pressure (MAP), (P = 0.012) and filtration gradient (GF), (P <0.001) were statistically significant between the first and second groups of patients in relation to the fourth. Differences in diuresis per hour (P = 0.022) showed statistical significance in relation to the first and third groups of patients.“

·       Additionally, the expression P =0.0001 should be changed with P<0.001 across the text.

·       Line 136 of the revised manuscript:  * symbol should be explained in the legend of the Table 2.

·       Line 144: in Table 3 statistical significance 0.001 should be written as exact number or as expression “<0.001”.

·       Line 144 (Table 3 and text lines 148-164) “R” (correlation coefficient) should be written as small letter (r).

Author Response

Dear reviewer, we apologize for not displaying the requested parameters (CT score), this is because some findings were not correct done. The distribution for the numerical variables was done and shown in the form of a normal distribution by Shapiro-Wilk and Kolmogorov-Smirnov test.

The number of patients by group was not written in Table 2 because it was previously stated in the text itself. Can the number of patients who participated in the study remain written in the text or must it be shown in Table 2? We are of the opinion that the table is more transparent in this way, but we will do as you suggested.

Zoran is a middle name, but it has now been corrected, i.e. deleted.

The results section in the abstract has been corrected as suggested.

The expression P =0.0001 has been changed with P<0.001 across the text.

* symbol is explained in the legend of table 2, represents the existence of statistical significance.

Statistical significance of 0.001 is written as < 0.001

The mark R is written with a lowercase letter r

Reviewer 3 Report

The manuscript has been updated to a much better version. It is good to go.

Author Response

Dear reviewer, the attached text has been modified according to the minor suggestions of another reviewer. Thank you for your comment.